# Reconfigurable droplet networks

Shuyi Sun[1], Shuailong Li[1], Weixiao Feng[1], Jiaqiu Luo[1], Thomas P. Russell ●[2,3] ✉ & Shaowei Shi ●[1] ✉

Droplet networks stabilized by lipid interfacial bilayers or colloidal particles have been extensively investigated in recent years and are of great interest for compartmentalized reactions and biological functions. However, current design strategies are disadvantaged by complex preparations and limited droplet size. Here, by using the assembly and jamming of cucurbit[8]uril surfactants at the oil-water interface, we show a novel means of preparing droplet networks that are multi-responsive, reconfigurable, and internally connected over macroscopic distances. Openings between the droplets enable the exchange of matter, affording a platform for chemical reactions and material synthesis. Our work requires only a manual compression to construct complex patterns of droplet networks, underscoring the simplicity of this strategy and the range of potential applications.

The formation of droplet networks typically relies on droplet interfacial bilayer (DIB), where the response and behavior of the lipids comprising the DIB are key[1]. When two droplets, stabilized by a monolayer of lipid surfactants, are brought into contact with one another, a DIB forms, linking the droplets. The DIB affords a pathway for the exchange of specific molecules between the droplets while, aside from the flat contact area of the DIB, the droplets maintain their original shape (Fig. 1a). Concatenated strings of droplets or more complex assemblies of droplets can be produced leading to the generation of cell-like networks stabilized by the DIBs[2]. Recently, the fabrication of controllable 1D, 2D, and 3D droplet patterns and microreactors based on droplet networks for mimicking an intracellular environment has gained significant attention[3,4]. Such systems typically involve the incorporation of biomolecules or functional materials, so as to realize a responsiveness to an electric field[5], pH[6], UV light[7], or other active transport species, as for example biological enzymes[8]. Such responsiveness has enabled the development of sensors, capacitors, and biological batteries[9–11], and systems for dye transmission, cargo delivery, cascade reactions in microreactors, and even the simulation of cell behavior and some smart soft materials[12–16]. Other droplet networks that do not involve DIBs, such as emulsion gels, produced by the bridging of droplets, are also sensitive to external stimuli, but the design of such droplet networks cannot be controlled and are, for the most part, random[17–19].

The maintenance of droplet shape is typically reliant on the use of surfactants. In comparison to conventional surfactants, colloidal particles have particular advantages, including the low concentration of particles required, low toxicity, minimal environmental impact, and the inherent functionality of the particles[20,21]. However, droplets stabilized by colloidal particles cannot form droplet networks, since the adsorption of the particles to the interface is irreversible and the areal density of the particles at the interface is very high[22,23]. By locally disrupting the interfacial assemblies with an external field, e.g., heating with a laser, the droplets can, though, be "welded" together[24]. Alternatively, the concentration of particles dispersed in the liquid can be reduced to a point where the interface is not saturated with particles and when two droplets are brought into contact, the droplets will coalesce and, as the interfacial area is reduced, the particles will jam, arresting the coalescence[25–27]. These procedures, however, require extremely precise control and manipulation with considerable energy consumption, making them difficult to implement.

Here, we demonstrate novel droplet networks based on cucurbit[8]uril (CB[8]) surfactants having an interpenetrating cavity that enables the interchange of materials and serves as a platform for compartmentalized chemical reactions (Fig. 1b). Unlike other droplet networks, the droplets used in this system can be macroscopic, several µL in volume, and require only a simple manual compression to affect, without the need of microfluidic devices or energy intensive

---

[1]State Key Laboratory of Chemical Resource Engineering, Beijing Advanced Innovation Center for Soft Matter Science and Engineering, Beijing University of Chemical Technology, 100029 Beijing, China. [2]Department of Polymer Science and Engineering, University of Massachusetts, Amherst, MA 01003, USA. [3]Materials Sciences Division, Lawrence Berkeley National Laboratory, 1 Cyclotron Road, Berkeley, CA 94720, USA. ✉e-mail: russell@mail.pse.umass.edu; shisw@mail.buct.edu.cn

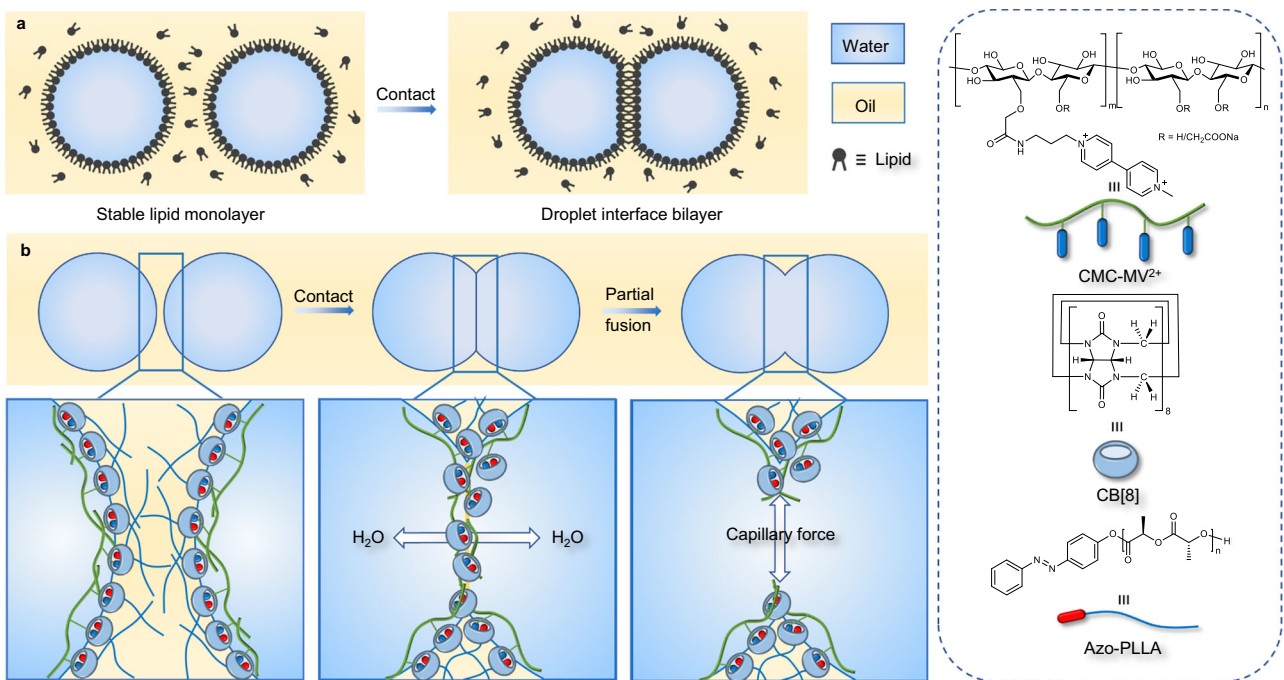

**Fig. 1 | Background introduction and our strategy for preparing droplet networks.** Schematics showing the formation of (**a**) droplet networks by DIB technique and (**b**) interpenetrating droplet networks by CB[8] surfactants.

processing. Internal penetration and direct material exchange within the droplet networks have been realized, rather than relying on the formation of a DIB or special proteins. The stability of the droplet network shape is directly attributable to the strength of the interfacial CB[8] surfactant assembly, rather than the subtle control of osmotic pressure or the introduction of a high viscosity fluid[28,29]. The droplet networks described here affords diversity in materials and operability of structured liquid droplet patterns and provides a simple platform for the preparation of structured, interconnected, and multi-responsive "soft" liquid microdevices.

## Results and discussion

### The formation, assembly, and jamming of CB[8] surfactants at the oil-water interface

To an aqueous solution of methyl viologen functionalized sodium carboxymethylcellulose (CMC-MV$^{2+}$, the amount of viologen moieties in CMC-MV$^{2+}$ is determined to be 0.62 mmol/g, pH = 8.0, Supplementary Scheme 1 and Supplementary Figs. 1–3), CB[8] is added to form a stable CMC-MV$^{2+}$⊂CB[8] binary complex (Supplementary Fig. 4). Azobenzene-terminated poly-L-polylactic acid (Azo-PLLA, $M_n$ = 5.2 K, Scheme S2 and Supplementary Figs. 5–6) is dissolved in an oil phase (toluene). At the interface between the aqueous and oil phases, CB[8] surfactants are generated and assemble in situ at the interface due to the high binding constant between the host CB[8] and the other two guest molecules, MV$^{2+}$ and Azo[30]. The kinetics of CB[8] surfactant formation and assembly at the interface is measured by tracking of the dynamic interfacial tension (IFT) using pendant drop tensiometry. As shown in Fig. 2a, b, with CMC-MV$^{2+}$⊂CB[8] complex dissolved in water against pure toluene, the IFT is almost the same as that of pure water with pure toluene (~36 mN m$^{-1}$), indicating no interfacial activity of the binary complexes, since CMC is very hydrophilic. On the other hand, with Azo-PLLA dissolved in toluene against pure water, the IFT is ~25 mN m$^{-1}$, indicating the surfactant nature of the ligand. When CMC-MV$^{2+}$⊂CB[8] complex dissolved in the water is placed against Azo-PLLA dissolved in toluene, the IFT rapidly decreases to ~17 mN m$^{-1}$, and wrinkles are observed on the droplet surface with only a very small compression (assembly time: 600 s; surface coverage: ~ 100%). These

wrinkles do not relax after 30 min (Fig. 2c, Supplementary Movie 1), indicating the formation, assembly and jamming of CB[8] surfactants at the interface, due to the high binding energy of the 1:1:1 MV$^{2+}$-Azo-CB[8] ternary complexes. Without the addition of host or guest molecules, no wrinkles are observed (Supplementary Fig. 7). During an extraction-reinjection process, the interfacial film contorts into unusual shapes, but returns to the initial droplet shape, indicating the robust nature of the assembled film (Fig. 2d and Supplementary Movie 2). The concentrations in each phase was decreased and the rheological properties of CB[8] surfactant-based interfacial assemblies were investigated[31]. The elastic (E'(ω)) component is always higher than the viscous (E"(ω)) component, demonstrating that the viscoelastic nature of the interfacial assembly is dominated by E'(ω) (Fig. 2e and Supplementary Fig. 8). 2D films prepared at a planar oil-water interface were transferred to a grid or silicon substrate for transmission electron microscope (TEM) and atomic force microscopy (AFM) studies. Uniform assemblies with smooth surfaces and an average thickness of ~14 nm were found (Fig. 2f, g and Supplementary Figs. 9 and 10).

### Multiple responsiveness of CB[8] surfactants

Since azobenzene can undergo a *trans*-to-*cis* isomerization upon exposure to UV light, the large steric hindrance of *cis*-azobenzene enables a rejection of Azo from the cavity of CB[8][32], leading to a disassembly of the ternary complexes at the interface. As shown in Fig. 3a, b, when the volume of the droplet is decreased to jam and wrinkles the interfacial assembly, upon exposure to UV light, the wrinkles disappear within 5 min, and the droplet returns to the classic spherical shape. When the UV light is turned off and the assembly is exposed to visible light, a slight wrinkling occurs, indicating the reassembly of CB[8] surfactants at the interface. On the other hand, if the guest molecule MV$^{2+}$ is initially reduced to MV$^{+}$ by Na$_2$S$_2$O$_4$, (MV$^{+}$)$_2$⊂CB[8], ternary complexes form in the aqueous phase (Fig. 3a)[33]. Against a toluene solution of Azo-PLLA in toluene, the IFT increases to ~25 mN m$^{-1}$, much higher than that originally, and no wrinkles are observed when the volume of the droplet is reduced, indicating that CB[8] surfactants do not form at the interface (Fig. 3c, d). When CMC-

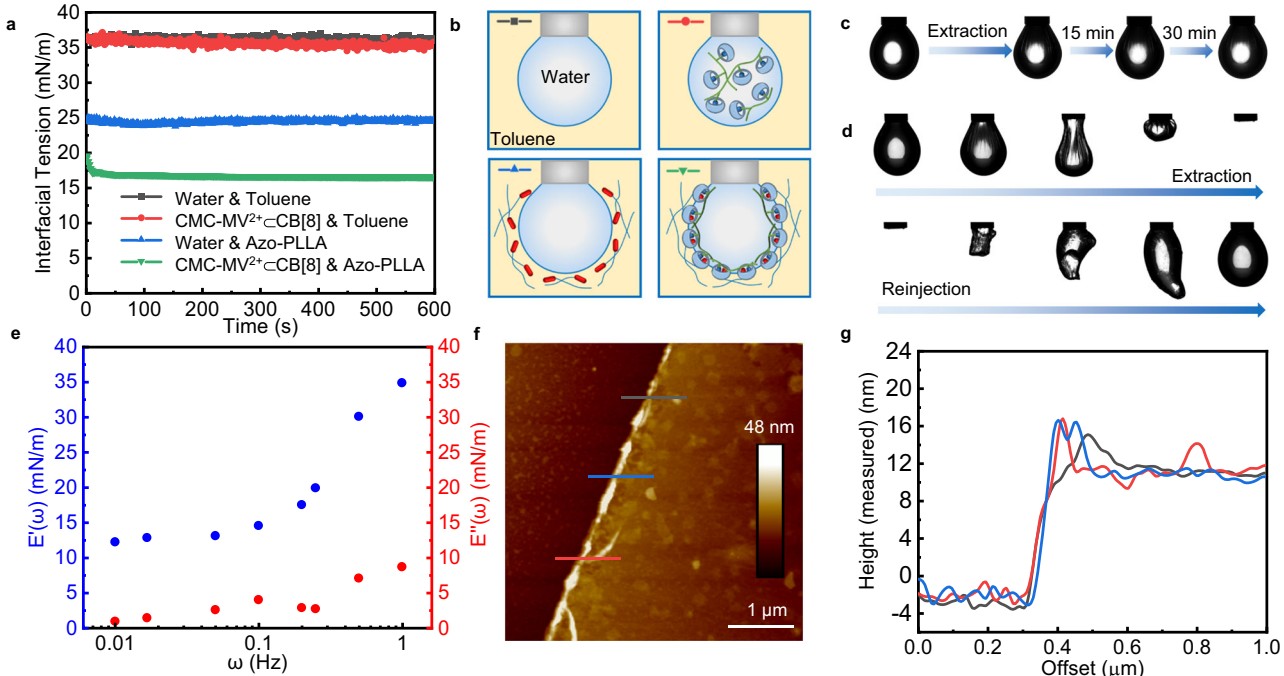

**Fig. 2 | The formation, assembly, and jamming of CB[8] surfactants at the oil-water interface. a** Time evolution of IFT for water & toluene, CMC-MV²⁺⊂CB[8] & toluene, water & Azo-PLLA, and CMC-MV²⁺⊂CB[8] & Azo-PLLA. **b** Schematic representation of **a**. **c** Morphology evolution of the pendent droplet with wrinkles on the surface. **d** A series of snapshots showing the shape evolution of a pendent droplet in an extraction–reinjection process. From **a**–**d**: [CMC-MV²⁺] = 0.5 mg mL⁻¹, [CB[8]] = 0.25 mg mL⁻¹, [Azo-PLLA] = 0.5 mg mL⁻¹. **e** Storage and loss dilatational moduli of CB[8] surfactant-based interfacial assemblies; [CMC-MV²⁺] = 0.3 mg mL⁻¹, [CB[8]] = 0.15 mg mL⁻¹, [Azo-PLLA] = 0.5 mg mL⁻¹, ω = 0.01–1 Hz. **f** AFM height image of the 2D film transferred from the interface and (**g**) line cut analysis of the film edge on the silicon substrate; [CMC-MV²⁺] = 0.5 mg mL⁻¹, [CB[8]] = 0.25 mg mL⁻¹, [Azo-PLLA] = 0.5 mg mL⁻¹.

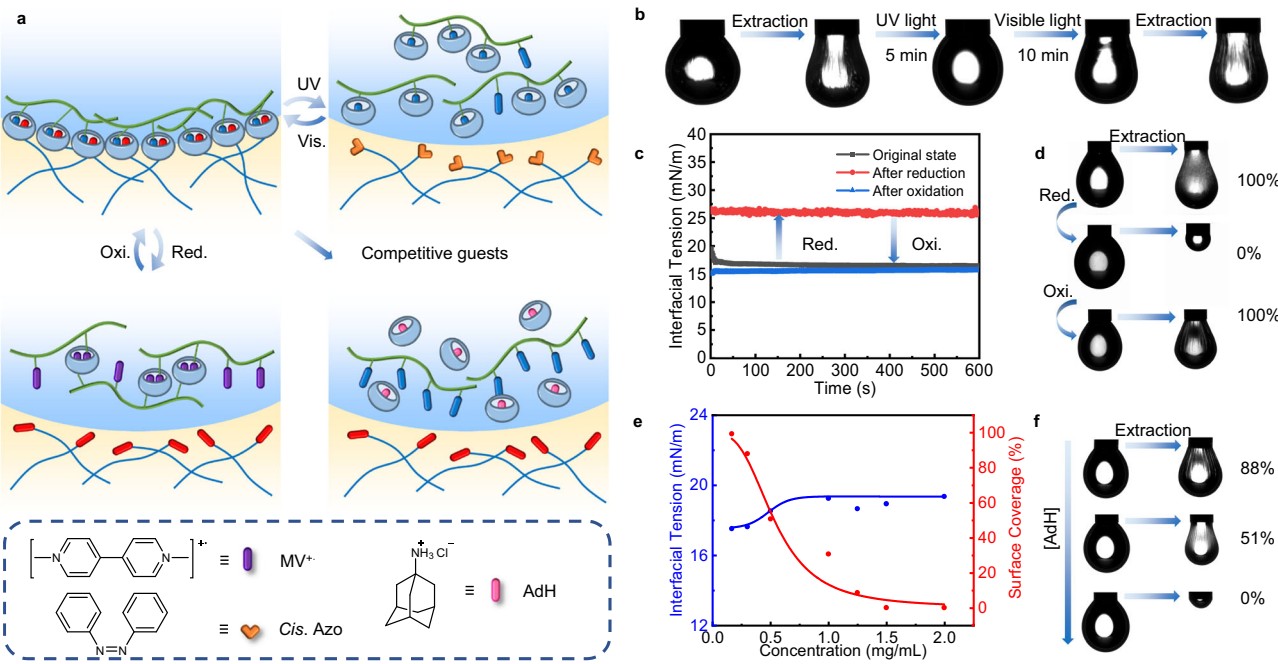

**Fig. 3 | Multiple responsiveness of CB[8] surfactants. a** Schematics showing the assembly and disassembly of CB[8] surfactants at the oil-water interface. **b** Morphology evolution of the pendent droplet under UV light or visible light. **c** Time evolution of IFT in a redox process. **d** Snapshots of droplet's morphology upon compression in a redox process. **e** Equilibrium IFT and surface coverage with the introduction of AdH in the aqueous phase. **f** Snapshots of droplet's morphology upon compression at different concentrations of AdH. [CMC-MV²⁺] = 0.5 mg mL⁻¹, [CB[8]] = 0.25 mg mL⁻¹, [Azo-PLLA] = 0.5 mg mL⁻¹.

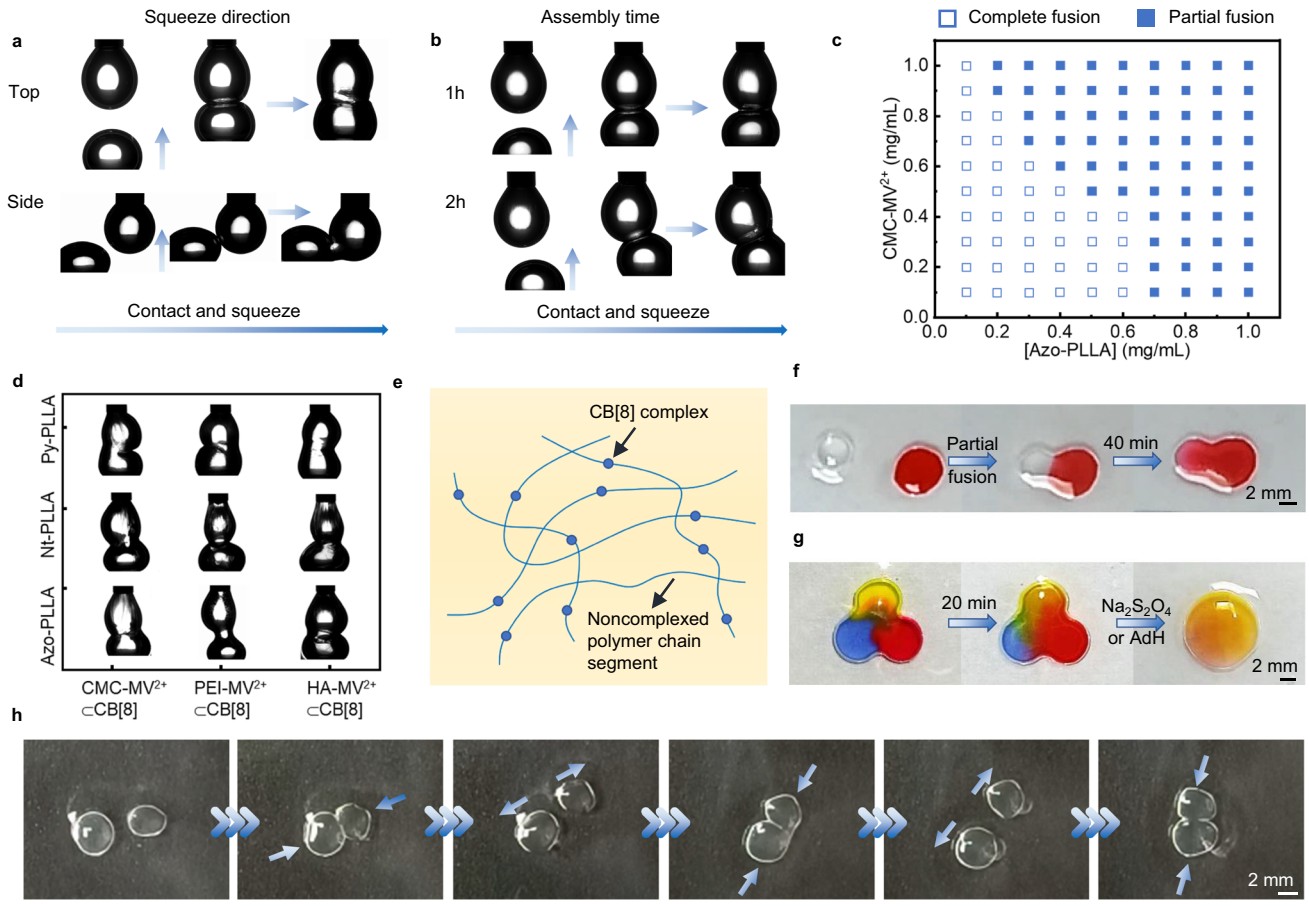

**Fig. 4 | Interpenetrating droplet networks stabilized by CB[8] surfactants.** Optical images showing the process of contact and squeeze of two droplets (**a**) in different direction and (**b**) at different assembly time; In all above cases, the partial fusion behavior of two droplets can be observed; [CMC-MV$^{2+}$] = 0.5 mg mL$^{-1}$, [CB[8]] = 0.25 mg mL$^{-1}$, [Azo-PLLA] = 0.5 mg mL$^{-1}$. **c** The effect of concentrations of CMC-MV$^{2+}$, CB[8] and Azo-PLLA ([CMC-MV$^{2+}$]: [CB[8]] = 2: 1) on arresting the coalescence of droplets (hollow symbols: complete fusion; solid symbols: partial fusion). **d** Optical images showing the partial fusion of droplets by using different CB[8] surfactant systems; [CMC-MV$^{2+}$] = 0.5 mg mL$^{-1}$, [PEI-MV$^{2+}$] = 0.5 mg mL$^{-1}$, [HA-

MV$^{2+}$] = 0.5 mg mL$^{-1}$, [CB[8]] = 0.25 mg mL$^{-1}$, [Azo-PLLA] = 0.5 mg mL$^{-1}$, [Nt-PLLA] = 0.5 mg mL$^{-1}$, [Py-PLLA] = 0.5 mg mL$^{-1}$. **e** Schematics showing polymer networks at the oil-water interface. **f** The diffusion of dye within the droplet pair; [Amaranth] = 0.5 mg mL$^{-1}$. **g** The mixture of primary colors within the droplet networks and the destruction of the droplet networks by adding reductant or competitive guest; [Sodium fluorescein] = 0.5 mg mL$^{-1}$, [Amaranth] = 0.5 mg mL$^{-1}$, [Nile blue A] = 0.5 mg mL$^{-1}$, [Na$_2$S$_2$O$_4$] = 2.0 mg mL$^{-1}$, [AdH] = 2.0 mg mL$^{-1}$. **h** Optical images of the formation-separation-reformation of droplet network for several times.

MV$^{+}$ is oxidized to CMC-MV$^{2+}$ by NaClO, the IFT decreases rapidly to ~17 mN m$^{-1}$, and the surface coverage saturates after 600 s, indicating the reformation and assembly of CB[8] surfactants. The degree of assembly and disassembly can, therefore, be controlled by varying the concentration of the oxidant or reductant (Supplementary Fig. 11). The in situ disassembly of CB[8] surfactants can be also achieved by adding Na$_2$S$_2$O$_4$ to an aqueous phase surrounding an oil droplet with an interfacial assembly of jammed CB[8] surfactants. Upon addition, the wrinkles on the surface rapidly disappear (Supplementary Fig. 12). Guest-competitive responsiveness can be achieved by adding a second guest that competes to complex with the CB[8]. One example is 1-adamantanamine hydrochloride (AdH) where the association constant of Ad with CB[8] is much higher than that of MV$^{2+}$(Fig. 3a)[34]. With the addition of AdH into the aqueous phase, CB[8] preferentially associates with AdH rather than MV$^{2+}$, leading to an increase in the IFT and a decrease in the surface coverage (Fig. 3e, f). Similarly, the in situ disassembly of CB[8] surfactants can be achieved by the addition of AdH to an inverted droplet, where wrinkles disappear and the droplet returns to a classic spherical shape in 40 s, as shown in Supplementary Fig. 13. Taking advantage of the dynamic nature of CB[8] surfactants, it is possible to prepare smart interfacial assemblies, e.g., microcapsules, with applications in many areas of materials and biological sciences, such as drug delivery, encapsulation, and controlled release[35–37].

## Interpenetrating droplet networks stabilized by CB[8] surfactants

By bringing two droplets with apparent 100% surface coverage of CB[8] surfactants into contact, surprisingly, at the contact position, a partial fusion is observed, but the remainder of the droplets remains in their original shape (Fig. 4a). To avoid gravitational effects, contact is made horizontally and, again, a partial fusion is observed. This partial fusion behavior can be observed once the surface coverage of CB[8] surfactants reaches to 100% (the threshold time for partial fusion can be estimated at 600 s at the fixed concentrations of CMC-MV$^{2+}$, CB[8] and Azo-PLLA shown in Fig. 4a, b), and after that the behavior is independent of the assembly time (Fig. 4b and Supplementary Fig. 14). This result also indicates that the structure of interfacial assembly remains unchanged once CB[8] surfactants saturate the interface, which is in agreement with the AFM results in Supplementary Fig. 10. By varying the concentration of CMC-MV$^{2+}$, CB[8] and Azo-PLLA, the threshold time can be effectively tuned (Supplementary Fig. 15). Similarly, the partial fusion behavior is independent of the concentration of CMC-MV$^{2+}$, CB[8] and Azo-PLLA, as shown in Fig. 4c, with the exception of some low concentrations, where the droplets fully fuse, due to the low surface coverage. By increasing the impact velocity of two droplet, a shorter time is needed to trigger the partial fusion of droplets (Supplementary Fig. 16). Moreover, three different water-

soluble $MV^{2+}$ functionalized polymers, CMC-$MV^{2+}$, PEI-$MV^{2+}$, and HA-$MV^{2+}$ (Scheme S3-S4, Supplementary Figs. 17–19), and three different oil soluble-polymer ligands, Azo-PLLA, naphthalene-terminated poly-L-polylactic acid (Nt-PLLA), and pyrene-terminated poly-L-polylactic acid (Py-PLLA; Supplementary Figs. 20–23), are used in the droplet contact experiments, and similar partial fusions are observed, as shown in Fig. 4d, showing the generality of the CB[8] surfactants in arresting the coalescence of droplets.

The droplets are encased in essentially a 2D network comprised of polymer chains with multiple CMCs that interact with multiple CB[8] surfactants on the surface that serve as permanent anchors to the interface, due to the high binding energy between the host and guest molecules and the rigidity of CB[8] itself. During the initial adsorption of the polymer chains to the surface, multiple points along the polymer chain are irreversibly anchored to the interface. As more polymer chains adsorb, they will bridge over the already adsorbed chains, anchoring to the interface at the unoccupied areas of the surface (Fig. 4e). Since the adsorption is irreversible, the anchoring points on one polymer can slide along other polymer chains but they cannot pass over or through the chains. Consequently, the anchoring points to the interface are analogous to junction points in a 3D network. If the network is deformed, the polymer chains can slide along each other until the chain segments between anchoring point fully stretched to relax the in-plane stress. We note that the anchoring points can also move along the interface as well. This allows the open areas to expand or contract in response to an applied in-plane forces. As illustrated in Fig. 1b, when two droplets are brought into contact, a bridging between the water in the two different droplets forms and will rapidly expand due to capillary forces[38], forcing the opening between the two droplets to enlarge. This provides easy pathways for cargo transport and chemical reactions based on the droplet pairs or networks. What initially limits the size of the channel between the droplets will be the extent to which the network can be stretched and disentangled on the surface. However, there is a continuous drive to decrease the interfacial area of the two droplets once the channels are established to coalesce the droplets. However, only a partial coalescence is observed where the merging of the droplets is arrested. The origins of this are as follows. As the opening between the droplets increases and once the network chains are fully stretched and then disentangled, an increasing number of CB[8] surfactants will have to move to open sites elsewhere on the surface. The redistribution of the CB[8] surfactants on the surface is such that the areal density of the CB[8] surfactants outside of the opening channel between the droplets increases. Eventually, the areal density will reach a point where the CB[8] surfactants will jam, arresting any further change in the shape of the merged droplets. This does open the very interesting possibility to be able to control the opening between the droplet, certainly on the microscopic and macroscopic length scales and opens the questions as to whether this can be extended to the nanoscopic level, an area we are currently investigating.

As shown in Fig. 4f and Supplementary Fig. 24, when two individual droplets (5-8 μL) are brought into contact (one with dye and one without), a dumbbell-like droplet pair are obtained. The diffusion of the dye is evident, since the entire droplet pair turns to red. By converting the color images to grayscale images and plotting the droplet profiles with gray values, time evolution of the diffusion process can be quantified (Supplementary Figs. 25 and 26). Similar results are obtained with a fluorescent dye using fluorescence microscopy. At the moment of droplet contact, the fluorescent dye diffuses rapidly and spreads throughout the droplet within 700 s (Supplementary Movie 3). The fluorescence intensity measured at a fixed position increases with time, then saturates at longer times (Supplementary Fig. 27). With the droplet volume ranging from 2.5 to 20 μL, the partial fusion can always be observed, indicating this behavior is independent of the droplet volume (Supplementary Fig. 28). When three droplets, dyed with colors of red, yellow, and blue, are brought into contact (Fig. 4g). Due to the dynamic nature of CB[8] surfactants, the full coalescence of droplets can be triggered by injecting a small amount of aqueous solution containing a reductant ($Na_2S_2O_4$) or competitive guest (AdH) into the droplets. As discussed above, CB[8] surfactants disassemble at the interface, significantly reducing the binding energy of interfacial assembly and, driven by the reduction in the interfacial tension, the triangular droplet networks rapidly become spherical, losing its structured shape. Furthermore, when two needles are used to slice the droplet pair, two independent droplets are produced that can be brought into contact and rejoined. This breaking and reformation process can be repeated multiple times, indicating the reconfigurability of the droplet networks (Fig. 4h and Supplementary Movie 4).

## Microreactors based on droplet networks
Droplet networks can also be served as microreactors. For example, when one droplet containing ferric chloride ($FeCl_3$) is brought into contact with a second droplet containing potassium thiocyanate (KSCN), the contact position of the two colorless droplets rapidly turns to blood red after the formation of droplet pairs, and the color change of the whole microreactor is realized within 10 min, indicating the production of ferric thiocyanate ($Fe(SCN)_3$, Fig. 5a, b, Supplementary Fig. 29, and Supplementary Movie 5). When one droplet contains cobaltous chloride ($CoCl_2$) and the other contains 2-methylimidazole (2-MI), microcrystals of zeolitic imidazole framework 67 (ZIF-67) can be synthesized within the droplet network, turning the droplets from colorless to purple (Fig. 5c, d, Supplementary Figs. 30 and 31, and Supplementary Movie 6)[39]. ZIF-8 can also be synthesized in a snowman-type of droplet pattern (Supplementary Fig. 32)[40]. Further, cascade reactions can be realized. As shown in Fig. 5e, f and Supplementary Fig. 33, droplet A (containing glucose) forms droplet pair with droplet B (containing horseradish peroxidase (HRP) and glucose oxidase (GOD)). Due to the reaction of glucose with GOD, hydrogen peroxide ($H_2O_2$) is produced[41]. At this time, droplet C (containing o-phenylenediamine (OPDA)) makes contact with droplet B. Due to the presence of $H_2O_2$, OPDA is oxidized to 2,3-diaminophenazine (DAP) and a yellow-green substance is produced in the droplets (Supplementary Movie 7).

## Complex patterns of droplet networks
Other interesting patterns based on droplet networks have also been constructed. For example, the basic building blocks of Tetris made from liquid droplets, such as type "I", type "L", etc (Fig. 6a, b). At the same time, the linear channel made from the droplet networks has realized the transmission of dyes (Fig. 6c, Supplementary Fig. 34, and Supplementary Movie 8), which is the biggest difference from the traditional droplet networks, proving that its internal structure is indeed interpenetrating. More complex droplet network channels can also be prepared (Supplementary Fig. 35). As shown in Fig. 6d, e, the transmission of three different colors of dye can be realized in the Y-shaped channels (Supplementary Movie 9). After transmission, the droplet channels still maintain their original form and are very stable.

In summary, we have designed a supramolecular polymer network at the water-oil interface based on the host-guest interaction of CMC-$MV^{2+}$⊂CB[8] and Azo-PLLA. When droplets stabilized by the supramolecular polymer network are brought into contact with each, water molecules can exchange rapidly through molecular channels that rapidly broaden due to capillary forces. The macroscopic manifestation is the partial fusion of droplets and the formation of a droplet network. However, due to the high bonding constant between the CB[8] and the other guest molecules, and the relatively strong mechanical strength of the interfacial film, the channels cannot broaden indefinitely and a restricted fusion is observed, maintaining the basic structure of the two droplets. Due to the multiple

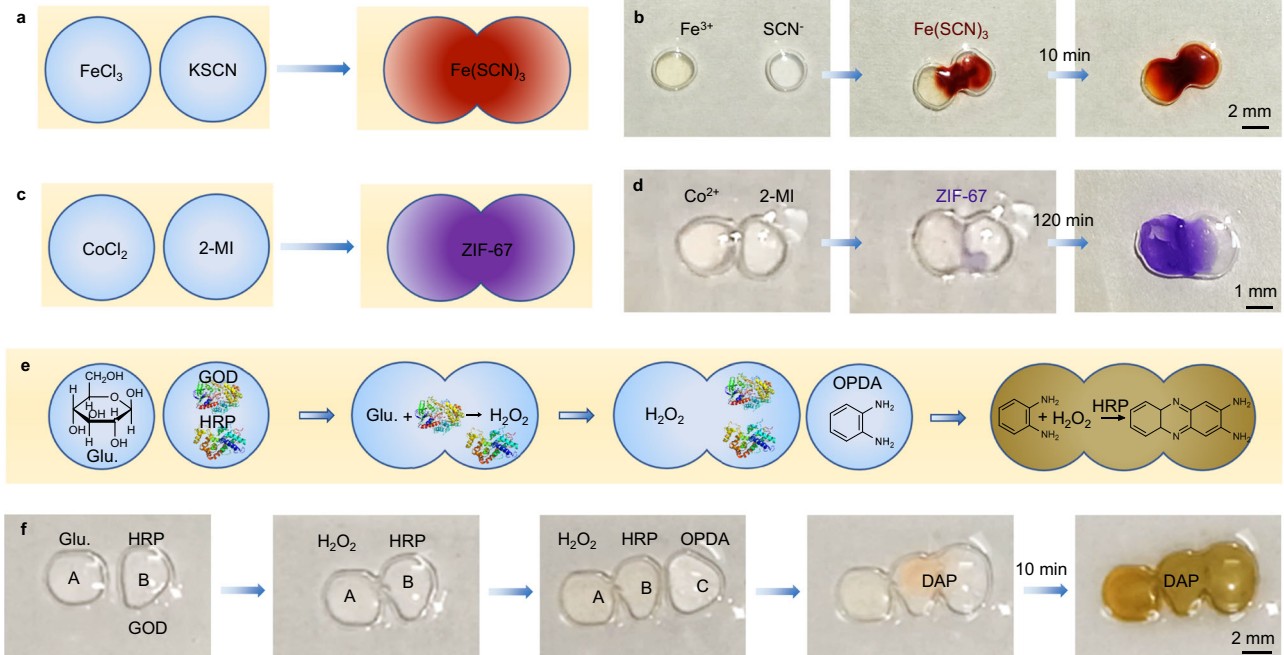

**Fig. 5 | Microreactors based on droplet networks. a, b** Schematics and optical images showing the chromogenic reaction for $Fe^{3+} + 3SCN^- = Fe(SCN)_3$ within the droplet networks. **c, d** Schematics and optical images showing the synthesis of ZIF-67 within the droplet networks. **e, f** Schematics and optical images showing the cascade enzymatic reaction within the droplet networks. $[CMC-MV^{2+}] = 1.0 \text{ mg mL}^{-1}$,

$[CB[8]] = 0.5 \text{ mg mL}^{-1}$, $[Azo\text{-}PLLA] = 1.0 \text{ mg mL}^{-1}$, $[Fe^{3+}] = 0.1 \text{ mg mL}^{-1}$, $[SCN^-] = 0.3 \text{ mg mL}^{-1}$, $[Co^{2+}] = 0.1 \text{ mg mL}^{-1}$, $[2\text{-MI}] = 0.5 \text{ mg mL}^{-1}$, $[Glucose] = 0.1 \text{ mg mL}^{-1}$, $[GOD] = 0.01 \text{ mg mL}^{-1}$, $[HRP] = 0.01 \text{ mg mL}^{-1}$, $[OPDA] = 0.1 \text{ mg mL}^{-1}$.

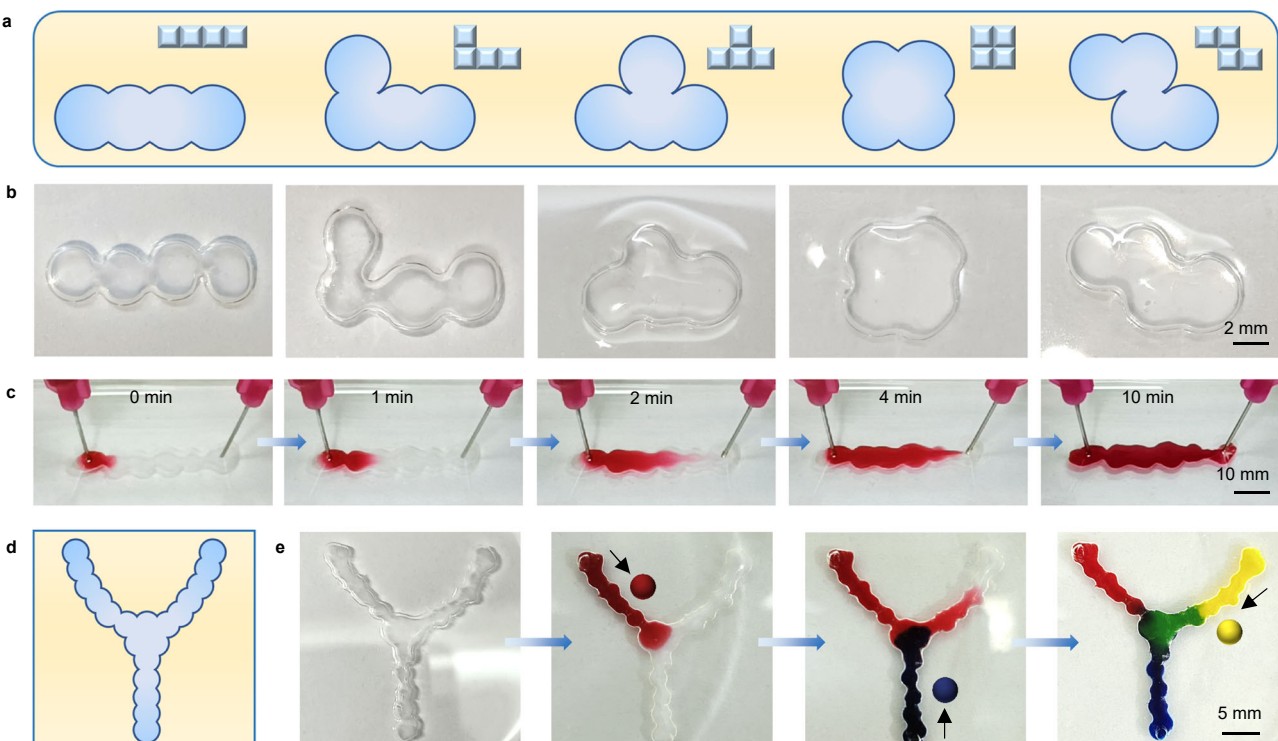

**Fig. 6 | Complex patterns of droplet networks. a, b** Schematics and optical images showing Tetris composed of droplet networks. **c** Optical images showing the transmission of a dye solution within the linear droplet network channel; $[Amaranth] = 1.0 \text{ mg mL}^{-1}$. **d** Schematic showing the "Y" type of droplet network

channels. **e** Optical images showing the transmission of three different kind of dye solutions within the "Y" type of droplet network channels; $[Amaranth] = 1.0 \text{ mg mL}^{-1}$, $[Nile \text{ blue A}] = 1.0 \text{ mg mL}^{-1}$, $[Sodium \text{ fluorescein}] = 1.0 \text{ mg mL}^{-1}$.

responsiveness of the guest molecules, applied external stimuli, such as reducing agents, and competitive guests, can be used to complete the total fusion of droplets. The droplet networks prepared by partial fusion can give the structured liquid patterns variety and operability, providing a simple path for the preparation of multiple responsive "soft" liquid microdevices.

## Methods

### Materials
4,4-bipyridine, iodomethane, 3-bromo-1-propanamine hydrobromide, bromoacetic acid, carboxymethylcellulose sodium salt (CMC, 1200 cps), Polyethyleneimine (PEI, $M_n = 10$ K), Hyaluronic acid (HA, $M_n = 100$ K), 1-ethyl-3-(3-dimethylaminopropyl) carbodiimide hydrochloride (EDC·HCl), N-hydroxysulfosuccinimide sodium salt (sulfo-NHS), MES (2-morpholinoethanesulfonic acid) buffer (0.2 M, pH = 5.5), 1-adamantanamine hydrochloride (AdH), ferric chloride (FeCl₃), potassium thiocyanate (KSCN), cobaltous chloride (CoCl₂·6H₂O), 2-methylimidazole (2-MI), Zinc Nitrate (Zn(NO₃)₂·6H₂O), horseradish peroxidase (HRP), glucose oxidase (GOD) and o-phenylenediamine (OPDA) were purchased from Macklin. ʟ-lactic acid (LLA), 4-Phenylazophenol, 2-Hydroxynaphthalene, 1-Pyrenol, Poly-ʟ lactic acid (PLLA, $M_n = 3.0$ K), and Stannous octoate were purchased from J&K. The above reagents were used as received unless otherwise noted. All anhydrous solvents (toluene, dichloromethane (DCM), acetonitrile (MeCN), tetrahydrofuran (THF), N, N-dimethylformamide (DMF), and ethanol) were purchased commercially and used without further purification.

### Instrumentation
The apparent number-average molecular weight ($M_n$) and dispersity ($M_w/M_n$) were measured by size-exclusion chromatography (SEC), which was conducted with THF as the eluent. Polymerization was monitored by ¹H NMR spectroscopy using a Bruker Advance 400 MHz NMR spectrometer with CDCl₃, D₂O, or DMSO-d₆ as solvent. Fourier transform infrared spectrum (FTIR) was recorded on a Spectrum One spectrophotometer by 64 scans from 4000 to 500 cm⁻¹ with a resolution of 4.0 cm⁻¹. The interfacial tension (γ) and interfacial dilatational rheology were analyzed by a multi-functional tensiometer (Krüss DSA30). The deformation and wrinkle behavior were recorded as images or videos with a digital camera. The morphology of droplet networks was characterized by polarized optical microscopy (ZEISS Imager.A2). 2D nanofilms prepared at the flat toluene-water interface were characterized by high-resolution transmission electron microscopy (Hitachi HT 7700) and atomic force microscope Bruker–Fastscan DMFASTSCAN2-SYS.

### Measurement of dynamic interfacial tension
The pendant drop method is used to measure the interfacial tension between water and toluene. The aqueous droplet containing CMC-MV²⁺⊂CB[8] is suspended vertically using a syringe needle into a glass cuvette containing toluene dissolving Azo-PLLA. The tensiometer provides the time evolution of interfacial tension γ by fitting a droplet's suspended shape to the Young–Laplace equation.

### Measurement of oscillatory pendant drop and rheometry
Interfacial dilatational rheology is performed using a pendant drop tensiometer (oscillating model). A fresh CMC-MV²⁺⊂CB[8]-containing water drop is created inside a toluene solution of Azo-PLLA, and before the start of testing, with drop volume fixed, interfacial assembly is allowed to reach equilibrium state. The rheological properties of the interfacial assembly are determined by an oscillatory dilation of the interfacial area, measuring the in-phase and out-of-phase component of the interfacial tension. The sinusoidal deformation of the surface area (ΔA/A₀) is kept at ~2% to remain within the linear viscoelastic regime. The interfacial tension and surface area of a droplet are measured as a function of time over the entire accessible frequency range (0.01–1 Hz). Different concentrations of CMC-MV²⁺, CB[8], and Azo-PLLA are used.

### Surface coverage calculation
For the calculation of surface coverage of CB[8] surfactant, a pendant droplet is first made by injecting a CMC-MV²⁺⊂CB[8]-containing aqueous solution at a given volume into Azo-PLLA-containing toluene solution. With time, CB[8] surfactants form at the toluene-water interface. At a certain assembly time, if CB[8] surfactants are uniformly distributed over the area of the interface (termed the "free" state), then as the volume and, therefore, the surface area of the pendant droplet is reduced, the CB[8] surfactants will become crowded at the interface, and eventually lose mobility, forming a "jammed" state at the interface. Wrinkling happens when CB[8] surfactants are closely jammed together at the interface. Therefore, the coverage ($C$) on the droplet surface in the "free" state could be estimated as $C \approx S_J/S_F$, where $S_J$ and $S_F$ are surface areas for the jammed and free states respectively, which can be achieved by tensiometer directly. By measuring the surface coverage at different assembly time, the evolution of surface coverage as a function of time can be achieved.

### Transmission of dye within the droplet network channels
To realize the continuous fluid transport in the droplet network channels, two stainless needles affixed to syringes attached to syringe pumps are inserted into the inlet and outlet of the channel, respectively. For experiments involving pumping amaranth solution through the single channel, the flow rate is fixed at 2.0 mL h⁻¹. For experiments involving pumping 3 dye solutions through the "Y" type of droplet network channels, the flow rate is fixed at 2.0 mL h⁻¹. The extraction needle is fixed at the intersection of three channels and the injection needle is fixed at the starting point of each channel.

## Data availability
The data that support the plots within this paper and other finding of this study are presented in the main article and the Supplementary Materials, the raw data are provided in Source Data file. All other additional data are available from the corresponding author upon request. Source data are provided with this paper.

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

## Acknowledgements

This work was supported by the National Natural Science Foundation of China (52322310, 52173018, 22372005) and Beijing Natural Science Foundation (2222071). TPR was supported by the U.S. Department of Energy, Office of Science, Office of Basic Energy Sciences, Materials Sciences and Engineering Division under Contract No. DE-AC02-05-CH11231 within the Adaptive Interfacial Assemblies Towards Structuring Liquids program (KCTR16).

## Author contributions

S.Sun, T.P.R. and S.Shi conceived the idea and designed the study. S.Sun carried out the pendant drop experiments, droplet network construction experiments, and droplet network reactor experiments. S.Sun, S.L., and J.L. performed synthesis and characterization of substances. S.Sun and W.F. performed the transmission of dye within the droplet network channel. S.Sun and S.Shi analyzed and interpreted the results. S.Sun, T.P.R., and S.Shi wrote the manuscript.

## Competing interests

The authors declare no competing interests.
