## [Peer Review File · Nature Communications]

REVIEWER COMMENTS

Reviewer #1 (Remarks to the Author):

In this manuscript the authors produce a novel emulsive system, creating elastic shells for microdroplets through the inclusion of nanoparticles in the aqueous phase and ligands in the surrounding oil phase. The authors begin by first describing the field of droplet-based materials through the DIB technique, then discuss results for monolayer tension measurements using pendant drop tensiometry with the varying presence of their surfactants and nanoparticles, clearly demonstrating the collaborative effect and elastic shells.

Next the authors show that this soft material exhibits an arrested coalescence, capable of forming diffusive pores between adjacent microdroplets for rapid mixing while partially preserving the original droplet shapes. Through a series of videos, the authors demonstrate various applications of the technology, and show relative ease in connecting and separating droplets.

The rate of diffusive exchange is made possible by the formation of large pores through capillary forces that are arrested by the elastic shells. This is a substantial advancement over the state of the art, and alleviates some of the issues with slower exchanges between droplets in DIBs. The manuscript examines multiple control cases, clearly defends the proposed mechanism, and presents new technology that will be interest to the general scientific community. Consequently, the manuscript merits publication. I do have a few comments and recommendations prior to publication to strengthen the manuscript:

- A primary concern is the length of time provided for arrested coalescence. The authors state that 2 hours was the observed threshold for consistent arrested coalescence and full collapse. Since the tension measurements indicate equilibrium was achieved after 10 minutes, is there a reason for this amount of time? A 2-hour delay between droplet formation and attachment will greatly hinder future applications of the technology. Is it possible to reduce this period of time?
- The manuscript details multiple responsiveness of the surfactants through UV exposure and a competitive guest effect. The competitive guest effect is used later in the manuscript to fully coalesce a three-droplet chain, and I do not think the UV exposure is utilized again. In this reviewer's opinion this section does not merit inclusion in the manuscript – while it does showcase other mechanics for the technology, applications for these mechanics are missing from the discussion and results.
- The references for the DIB introduction are largely from older manuscripts, and a few are questionably cited (ref 6 discusses a hair cell sensor but is discussed in the text as active species transport). I would recommend replacing several of the older references with more up to date discussions on DIB networks and materials.
- The methods section for characterization is limited. More information should be provided on the characterization of the viscoelastic properties of the shells, since this is a crucial aspect of the approach.

- Some discussion of the pore dimensions should be presented. Is it possible to tune the rate of diffusion by adjusting the droplet dimensions or adjusting the properties of the elastic shells? How reliable is the diffusion rate between experiments? Repeating the experiments in Fig 5 and plotting the intensity to monitor diffusion between the droplets and quantifying the expected deviation per experiment will greatly enhance the manuscript. If this technology is to be adapted for precise micromixers, then more characterization of the exchanges is necessary beyond the proof-of-concept demonstrations presented here.

Summary: Major Revisions

Reviewer #2 (Remarks to the Author):

The manuscript examines interfacially jammed droplets, the networks they form, the reactions that govern the formation of structures at their surfaces, and provide some preliminary results around the use of these droplets to create networks and carry out reactions within the zones created by the necks and the network. The work is of interest to a broad audience of soft matter physicists, chemical engineers studying complex liquids, and other chemists who work in the area of interfacial behaviour. There are some nice demonstrations of control of the reversibility of the interfacial reactions used to weld or arrest the droplets here.

The paper could eventually be published, but needs to increase the quantitative nature of its characterisation work. Right now there are a lot of demonstrated phenomena but not very much insight that will allow design and use of these materials and phenomena going forward. I recommend the work be revised to improve its insights. Some examples are provided below:

The work is mostly qualitative, without particularly quantitative analysis of the results so that future workers have a basis for design or use of these materials. For example, the arrest and jamming thresholds could be related to mechanical properties like modulus, or controllable parameters like surface concentration and reaction extent, but no attempt is made here.

One of the key properties of arrested systems like this is the elastic modulus. These values are plotted, for example, in Figure 2 of the manuscript and also a figure in the SI. Despite the centrality of these methods, no method is given for measuring the modulus so we are left without insight into the accuracy or relevance of these key values.

Reviewer #3 (Remarks to the Author):

In this manuscript the authors present a new generation of couplable droplets/vesicles. The topic is relevant since droplet based platforms represent an emerging trend. Although networks of interconnected droplets have been extensively researched over the past decade, the successful demonstration of reversible direct connectivity between droplets (partial inter-droplet fusion) rather than through a permeable membrane is exciting. The work also illustrates the operability and potential applications of the novel droplet network approach. Therefore, the authors have found a highly attractive phenomena that is worth reporting on a high standard journal as Nature Communications. However, at the current stage, the paper is not very clearly written (mainly the abstract) and the following comments should be addressed before publication.

1) The motivation of the current study is somehow unclear in the Title and Abstract. In my view, the key novelty of this paper is the idea of reconfigurable partially fused droplet networks (the title should depict this novelty). The concept of direct interconnectivity between droplets instead of membrane-mediated connection should be clearer in the Abstract. The Abstract should also be rewritten in a more standard format (initiate with the motivation of the work, followed by a phrase like "Here, we show" to begin to briefly summarize the main results and highlight the novelty and implications of the work.

2) It would be interesting for example to explore some range of parameters for which droplet coupling occurs, changing systematically the impact velocity or the layer thickness.

3) It would also have been interesting to get a more quantitative supported understanding of the partial fusion of the droplets.

4) A more detailed discussion on the redox responsiveness to control droplet fusion should be provided.

Manuscript ID: NCOMMS-23-36167-T

Title: Reconfigurable Droplet Networks

Authors: Shuyi Sun, Shuailong Li, Weixiao Feng, Jiaqiu Luo, Thomas P. Russell* and Shaowei Shi*

Dear editor:

We would like to thank the reviewers for their detailed feedback to improve our submission. We have considered each comment carefully and revised our manuscript to address the issues raised.

Responses to Reviewer #1

Comments:

In this manuscript the authors produce a novel emulsive system, creating elastic shells for microdroplets through the inclusion of nanoparticles in the aqueous phase and ligands in the surrounding oil phase. The authors begin by first describing the field of droplet-based materials through the DIB technique, then discuss results for monolayer tension measurements using pendant drop tensiometry with the varying presence of their surfactants and nanoparticles, clearly demonstrating the collaborative effect and elastic shells.

Next the authors show that this soft material exhibits an arrested coalescence, capable of forming diffusive pores between adjacent microdroplets for rapid mixing while partially preserving the original droplet shapes. Through a series of videos, the authors demonstrate various applications of the technology, and show relative ease in connecting and separating droplets.

The rate of diffusive exchange is made possible by the formation of large pores through capillary forces that are arrested by the elastic shells. This is a substantial advancement over the state of the art, and alleviates some of the issues with slower exchanges between droplets in DIBs. The manuscript examines multiple control cases, clearly defends the proposed mechanism, and presents new technology that will be interest to the general scientific community. Consequently, the manuscript merits publication. I do have a few comments and recommendations prior to publication to strengthen the manuscript:

1. A primary concern is the length of time provided for arrested coalescence. The authors state that 2 hours was the observed threshold for consistent arrested coalescence and full collapse. Since the tension measurements indicate equilibrium was achieved after 10 minutes, is there a reason for this amount of time? A 2-hour delay between droplet formation and attachment will greatly hinder future applications of the technology. Is it possible to reduce this period of time?

Response: We appreciate the comment from the reviewer and apologize for the incorrect statement “This partial fusion behavior was observed as long as the time allowed for the assembly of the CB[8] surfactants on the original droplets was greater

than 2 h”. In fact, the partial fusion behavior can be observed once the surface coverage of CB[8] surfactants reaches to $\sim 100\%$, that means the assembly time it takes to saturate the surface is the threshold for arrested coalescence. To measure the threshold value, the surface coverage (C) of CB[8] surfactants as a function of time was investigated, where $C \approx S_J/S_F$ and S_J and S_F are the surface areas of the jammed and free (initial) states (*Langmuir* **2013**, *29*, 13407). As shown in Fig.R1a, when the concentrations of CMC-MV²⁺, CB[8] and Azo-PLLA are fixed at 0.5, 0.25 and 0.5 mg mL⁻¹, the surface coverage reaches to $\sim 100\%$ in 600 s. Therefore, 600 s can be estimated as the threshold value for arrested coalescence and, when the assembly time is greater than 600 s, partial fusion behavior can be observed (Fig.R1b). We note that, the partial fusion behavior can also be observed when the assembly time is slightly less than 600 s, e.g., 400 s. However, in this case, the coalescence of two droplets cannot be fully arrested. By varying the concentration of CMC-MV²⁺, CB[8] or Azo-PLLA, the assembly time it takes to saturate the surface, i.e., threshold time can be effectively tuned (Fig.R2).

We have made a revision in the manuscript as follows: “This partial fusion behavior can be observed once the surface coverage of CB[8] surfactants reaches to 100% (the threshold time for partial fusion can be estimated at 600 s at the fixed concentrations of CMC-MV²⁺, CB[8] and Azo-PLLA shown in Fig.4a,b), and after that the behavior is independent of the assembly time (Fig.4b and Fig.S14). By varying the concentration of CMC-MV²⁺, CB[8] and Azo-PLLA, the threshold time can be effectively tuned (Fig.S15)”.

We have also added Fig.R1 and Fig.R2 in the revised supporting information as Fig.S14 and Fig.S15.

Fig.R1 (a) Evolution of the surface coverage as a function of time. (b) Optical images showing the process of contact and squeeze of two droplets at different assembly time; [CMC-MV²⁺] = 0.5 mg mL⁻¹, [CB[8]] = 0.25 mg mL⁻¹, [Azo-PLLA] = 0.5 mg mL⁻¹.

Fig.R2 (a) Evolution of the threshold time as a function of concentration of Azo-PLLA; $[CMC-MV^{2+}] = 0.5 \text{ mg mL}^{-1}$, $[CB[8]] = 0.25 \text{ mg mL}^{-1}$. (b) Evolution of the threshold time as a function of concentration of CMC-MV²⁺/CB[8]; $[CMC-MV^{2+}] : [CB[8]] = 2:1$, $[Azo-PLLA] = 0.5 \text{ mg mL}^{-1}$.

2. The manuscript details multiple responsiveness of the surfactants through UV exposure and a competitive guest effect. The competitive guest effect is used later in the manuscript to fully coalesce a three-droplet chain, and I do not think the UV exposure is utilized again. In this reviewer's opinion this section does not merit inclusion in the manuscript – while it does showcase other mechanics for the technology, applications for these mechanics are missing from the discussion and results.

Response: We appreciate the comment from the reviewer. As shown in the “Multiple responsiveness of CB[8] surfactants” section, CB[8] surfactants are independently responsive to the light, redox agent and competitive guest. Therefore, the coalescence of droplet networks can be triggered by anyone of these three stimuli (as the reviewer thought).

In this work, we mainly focused on the formation mechanism and application of a novel interfacial system, i.e., interconnecting droplet network, and did not give a detailed discussion on the applications related to the responsiveness of CB[8] surfactants. However, it is still worthwhile to study the dynamic nature of CB[8] surfactants, since it would offer the possibility to construct smart interfacial assemblies such as microcapsules and emulsions, that can be used in many fields such as drug delivery, encapsulation, and controlled release (*Science* **2012**, 335, 690; *Nat. Commun.* **2014**, 5, 5772; *Acc. Chem. Res.* **2017**, 50, 208). Thus, we prefer to keep this section in the manuscript.

To address the necessity and importance of the study on the responsiveness of CB[8] surfactants, we have added a brief discussion in the revised manuscript as follows: “Taking advantage of the dynamic nature of CB[8] surfactants, it is possible to prepare smart interfacial assemblies, e.g., microcapsules, with applications in many areas of materials and biological sciences, such as drug delivery, encapsulation, and controlled release (*Science* **2012**, 335, 690; *Nat. Commun.* **2014**, 5, 5772; *Acc. Chem. Res.* **2017**, 50, 208)”. Moreover, in Fig.5f, we have removed the picture related to the responsiveness behavior of droplet networks, since there is no direct relationship

between the cascade enzymatic reaction and the responsiveness behavior.

3. The references for the DIB introduction are largely from older manuscripts, and a few are questionably cited (ref 6 discusses a hair cell sensor but is discussed in the text as active species transport). I would recommend replacing several of the older references with more up to date discussions on DIB networks and materials.

Response: We appreciate the suggestion from the reviewer, and have modified the references (ref.5-16) for the DIB introduction.

4. The methods section for characterization is limited. More information should be provided on the characterization of the viscoelastic properties of the shells, since this is a crucial aspect of the approach.

Response: We appreciate the suggestion from the reviewer, and have added more experimental methods including measurement of dynamic interfacial tension, measurement of oscillatory pendant drop and rheometry and surface coverage calculation in the revised manuscript.

Measurement of dynamic interfacial tension

The pendant drop method is used to measure the interfacial tension between water and toluene. The aqueous droplet containing CMC-MV²⁺-CB[8] is suspended vertically using a syringe needle into a glass cuvette containing toluene dissolving Azo-PLLA. The tensiometer provides the time evolution of interfacial tension γ by fitting a droplet's suspended shape to the Young–Laplace equation.

Measurement of oscillatory pendant drop and rheometry

Interfacial dilatational rheology is performed using a pendant drop tensiometer (oscillating model). A fresh CMC-MV²⁺-CB[8]-containing water drop is created inside a toluene solution of Azo-PLLA, and before the start of testing, with drop volume fixed, interfacial assembly is allowed to reach equilibrium state. The rheological properties of the interfacial assembly are determined by an oscillatory dilation of the interfacial area, measuring the in-phase and out-of-phase component of the interfacial tension. The sinusoidal deformation of the surface area ($\Delta A/A_0$) is kept at $\sim 2\%$ to remain within the linear viscoelastic regime. The interfacial tension and surface area of a droplet are measured as a function of time over the entire accessible frequency range (0.01–1 Hz). Different concentrations of CMC-MV²⁺, CB[8] and Azo-PLLA are used.

Surface coverage calculation

For the calculation of surface coverage of CB[8] surfactant, a pendant droplet is first made by injecting a CMC-MV²⁺-CB[8]-containing aqueous solution at a given volume into Azo-PLLA-containing toluene solution. With time, CB[8] surfactants form at the toluene-water interface. At a certain assembly time, if CB[8] surfactants are uniformly distributed over the area of the interface (termed the “free” state), then as the volume and, therefore, the surface area of the pendant droplet is reduced, the CB[8] surfactants will become crowded at the interface, and eventually lose mobility, forming a “jammed” state at the interface. Wrinkling happens when CB[8] surfactants are closely jammed together at the interface. Therefore, the coverage (C) on the droplet surface in the “free” state could be estimated as $C \approx S_J/S_F$, where S_J and S_F are surface areas for the jammed

and free states respectively, which can be achieved by tensiometer directly. By measuring the surface coverage at different assembly time, the evolution of surface coverage as a function of time can be achieved.

5. Some discussion of the pore dimensions should be presented. Is it possible to tune the rate of diffusion by adjusting the droplet dimensions or adjusting the properties of the elastic shells? How reliable is the diffusion rate between experiments? Repeating the experiments in Fig 5 and plotting the intensity to monitor diffusion between the droplets and quantifying the expected deviation per experiment will greatly enhance the manuscript. If this technology is to be adapted for precise micromixers, then more characterization of the exchanges is necessary beyond the proof-of-concept demonstrations presented here.

Response: We appreciate the comments from the reviewer. In this work, since the droplet networks are made by a simple manual compression and it is hard to control the applied force, the pore dimension is difficult to be precisely tuned and discussed.

For the droplet networks stabilized by CB[8] surfactants, we note that, there is no membrane at the junction of droplets and macroscopic pore are generated (millimeter-scale). The nano-scale diffusants used in this work can freely diffuse from one droplet to another, and the rate of diffusion between experiments is not affected by the droplet dimensions or the properties of the elastic shells. The only parameter that can be affected is the flux. If the pore dimension between droplets is made larger and the concentration of components remains the same, then the number of diffusants passing the pore per unit time will increase, leading to an increased flux.

We also agree with the reviewer that more characterization of the exchanges is necessary if this technology is to be adapted for precise micromixers. However, our intent is not to demonstrate an actual device in use but, rather, to lay down the fundamental science underpinning the potential application in some fields.

Responses to Reviewer #2

Comments:

The manuscript examines interfacially jammed droplets, the networks they form, the reactions that govern the formation of structures at their surfaces, and provide some preliminary results around the use of these droplets to create networks and carry out reactions within the zones created by the necks and the network. The work is of interest to a broad audience of soft matter physicists, chemical engineers studying complex liquids, and other chemists who work in the area of interfacial behaviour. There are some nice demonstrations of control of the reversibility of the interfacial reactions used to weld or arrest the droplets here.

The paper could eventually be published, but needs to increase the quantitative nature of its characterisation work. Right now there are a lot of demonstrated phenomena but not very much insight that will allow design and use of these materials and phenomena going forward. I recommend the work be revised to improve its insights. Some examples are provided below:

1. The work is mostly qualitative, without particularly quantitative analysis of the results so that future workers have a basis for design or use of these materials. For example, the arrest and jamming thresholds could be related to mechanical properties like modulus, or controllable parameters like surface concentration and reaction extent, but no attempt is made here.

Response: We appreciate the suggestion from the reviewer. It should be noted that, the partial fusion behavior can only be observed after the surface coverage of CB[8] surfactants reaches to $\sim 100\%$. We agree with the reviewer that the mechanical properties of the interfacial assembly may play an important role in arresting the coalescence of droplets. However, after the surface coverage of CB[8] surfactants reaching to $\sim 100\%$, the thickness of the interfacial film measured by AFM remains unchanged (~ 14 nm), independent of the assembly time and concentration of CMC-MV²⁺, CB[8] and Azo-PLLA (Fig.R3), and the partial fusion behaviors are similar. These results demonstrate that, once CB[8] surfactants saturate the surface, the residues that are dissolved in water and oil phase will not continuously adsorb to the interface and interact with each other to thicken the film. Therefore, it is anticipated that the modulus of interfacial assembly also remains unchanged and it is difficult to quantitatively investigate the arresting and jamming thresholds by tuning the modulus.

Fig.R3 (a) Interfacial film thickness as the function of assembled time; [CMC-MV²⁺] = 0.5 mg mL⁻¹, [CB[8]] = 0.25 mg mL⁻¹ [Azo-PLLA] = 0.5 mg mL⁻¹. (b) Interfacial film thickness as the function of CMC-MV²⁺/CB[8] concentration; [Azo-PLLA] = 0.5 mg mL⁻¹, [CMC-MV²⁺] = 0.25-1.25 mL⁻¹, [CMC-MV²⁺] : [CB[8]] = 2:1, Assemble time > 600 s. (c) Interfacial film thickness as the function of Azo-PLLA concentration; [Azo-PLLA] = 0.25-1.25 mg mL⁻¹, [CMC-MV²⁺] = 0.5 mL⁻¹, [CB[8]] = 0.25 mg mL⁻¹, Assemble time > 600 s.

Here we provide another approach to quantitatively understand the partial fusion of the droplets. Since the partial fusion behavior can only be observed after the surface coverage of CB[8] surfactants reaches to $\sim 100\%$, the assembly time it takes to saturate the surface can be estimated as the threshold for arrested coalescence. To measure the threshold value, the surface coverage (C) of CB[8] surfactants as a function of time was investigated, where $C \approx S_J/S_F$ and S_J and S_F are the surface areas of the jammed and free (initial) states (*Langmuir* **2013**, *29*, 13407). As shown in Fig.R1a, when the concentrations of CMC-MV²⁺, CB[8] and Azo-PLLA are fixed at 0.5, 0.25 and 0.5 mg

mL⁻¹, the surface coverage reaches to ~ 100% in 600 s. Therefore, 600 s can be estimated as the threshold value for arrested coalescence and, when the assembly time is greater than 600 s, partial fusion behavior can be observed (Fig.R1b). We note that, the partial fusion behavior can also be observed when the assembly time is slightly less than 600 s, e.g., 400 s. However, in this case, the coalescence of two droplets cannot be fully arrested. By varying the concentration of CMC-MV²⁺, CB[8] or Azo-PLLA, the assembly time it takes to saturate the surface, i.e., threshold time can be effectively tuned (Fig.R2).

We have added a brief discussion in the manuscript as follows: “This partial fusion behavior can be observed once the surface coverage of CB[8] surfactants reaches to 100% (the threshold time for partial fusion can be estimated at 600 s at the fixed concentrations of CMC-MV²⁺, CB[8] and Azo-PLLA shown in Fig.4a,b), and after that the behavior is independent of the assembly time (Fig.4b and Fig.S14). By varying the concentration of CMC-MV²⁺, CB[8] and Azo-PLLA, the threshold time can be effectively tuned (Fig.S15)”.

We have also added Fig.R1-R3 in the revised supporting information as Fig.S14, Fig.S15 and Fig.S10.

2. One of the key properties of arrested systems like this is the elastic modulus. These values are plotted, for example, in Figure 2 of the manuscript and also a figure in the SI. Despite the centrality of these methods, no method is given for measuring the modulus so we are left without insight into the accuracy or relevance of these key values.

Response: We appreciate the suggestion from the reviewer, and have added the measurement of oscillatory pendant drop and rheometry in the revised manuscript.

Measurement of oscillatory pendant drop and rheometry

Interfacial dilatational rheology is performed using a pendant drop tensiometer (oscillating model). A fresh CMC-MV²⁺/CB[8]-containing water drop is created inside a toluene solution of Azo-PLLA, and before the start of testing, with drop volume fixed, interfacial assembly is allowed to reach equilibrium state. The rheological properties of the interfacial assembly are determined by an oscillatory dilation of the interfacial area, measuring the in-phase and out-of-phase component of the interfacial tension. The sinusoidal deformation of the surface area ($\Delta A/A_0$) is kept at ~ 2 % to remain within the linear viscoelastic regime. The interfacial tension and surface area of a droplet are measured as a function of time over the entire accessible frequency range (0.01–1 Hz). Different concentrations of CMC-MV²⁺, CB[8] and Azo-PLLA are used.

References related to the interfacial rheology: *Soft Matter* **2016**, *12*, 8701; *Soft Matter* **2011**, *7*, 7586; *Langmuir* **2012**, *28*, 8052; *Langmuir* **2017** *33*, 7994.

Responses to Reviewer #3

Comments:

In this manuscript the authors present a new generation of couplable droplets/vesicles. The topic is relevant since droplet based platforms represent an emerging trend. Although networks of interconnected droplets have been extensively researched over the past decade, the successful demonstration of reversible direct connectivity between

droplets (partial inter-droplet fusion) rather than through a permeable membrane is exciting. The work also illustrates the operability and potential applications of the novel droplet network approach. Therefore, the authors have found a highly attractive phenomenon that is worth reporting on a high standard journal as Nature Communications. However, at the current stage, the paper is not very clearly written (mainly the abstract) and the following comments should be addressed before publication.

1. The motivation of the current study is somehow unclear in the Title and Abstract. In my view, the key novelty of this paper is the idea of reconfigurable partially fused droplet networks (the title should depict this novelty). The concept of direct interconnectivity between droplets instead of membrane-mediated connection should be clearer in the Abstract. The Abstract should also be rewritten in a more standard format (initiate with the motivation of the work, followed by a phrase like “Here, we show” to begin to briefly summarize the main results and highlight the novelty and implications of the work).

Response: We appreciate the suggestion from the reviewer, and have revised the title to “Reconfigurable Droplet Networks”.

The abstract has been revised as follows: “Droplet networks stabilized by lipid interfacial bilayers or colloidal particles have been extensively investigated in recent years and are of great interest for compartmentalized reactions and biological functions. However, current design strategies are disadvantaged by complex preparations and limited droplet size. Here, by using the assembly and jamming of cucurbit[8]uril surfactants at the oil-water interface, we show a novel means of preparing droplet networks that are multi-responsive, reconfigurable, and internally connected over macroscopic distances. Opening between the droplets enables the exchange of matter, affording a platform for chemical reactions and material synthesis. Our work requires only a manual compression to construct complex patterns of droplet networks, underscoring the simplicity of this strategy and the range of potential applications”.

2. It would be interesting for example to explore some range of parameters for which droplet coupling occurs, changing systematically the impact velocity or the layer thickness.

Response: We appreciate the suggestion from the reviewer. As shown in Fig.R4, with fixed concentrations of CMC-MV²⁺, CB[8] and Azo-PLLA, when the upper droplet moves to the bottom droplet (pinned to the substrate) at different rates, it is found that, with the increasing impact velocity from 3.86 to 6.36 mm/s, the time for opening the pore decreases from 0.12 to 0.017 s. Since the impact velocity is proportional to the compressive force, these results demonstrate that it is easier to trigger the partial fusion of droplets by larger compressive forces.

We have added a brief discussion in the revised manuscript as follows: “By increasing the impact velocity of two droplets, a shorter time is needed to trigger the partial fusion of droplets (Fig.S16)” and added Fig.R4 in the revised supporting information as Fig.S16.

Fig.R4 Optical images showing the effect of impact velocity on triggering the partial fusion of two droplets. $[\text{CMC-MV}^{2+}] = 0.5 \text{ mg mL}^{-1}$, $[\text{CB}[8]] = 0.25 \text{ mg mL}^{-1}$, $[\text{Azo-PLLA}] = 0.5 \text{ mg mL}^{-1}$.

We also tried to investigate the effect of layer thickness on triggering the partial fusion of droplets. However, it is found that, after the surface coverage of CB[8] surfactants reaching to $\sim 100\%$ (the precondition for achieving partial fusion), the thickness of the interfacial film remains unchanged ($\sim 14 \text{ nm}$), independent of the assembly time and concentration of CMC-MV²⁺, CB[8] and Azo-PLLA (Fig. R3), and the partial fusion behaviors are similar. These results demonstrate that, once CB[8] surfactants saturate the surface, the residues that are dissolved in water and oil phase will not continuously adsorb to the interface and interact with each other to thicken the film. Thus, it is difficult to systematically vary the thickness of the interfacial film.

We have added Fig.R3 in the revised supporting information as Fig.S10.

3. It would also have been interesting to get a more quantitative supported understanding of the partial fusion of the droplets.

Response: We appreciate the suggestion from the reviewer. To quantitatively understanding the partial fusion of the droplets, the threshold for arrested coalescence is investigated. It should be noted that, the partial fusion behavior can be observed once the surface coverage of CB[8] surfactants reaches to $\sim 100\%$, that means the assembly time it takes to saturate the surface is the threshold for arrested coalescence. To measure the threshold value, the surface coverage (C) of CB[8] surfactants as a function of time was investigated, where $C \approx S_J/S_F$ and S_J and S_F are the surface areas of the jammed and free (initial) states (*Langmuir* **2013**, *29*, 13407). As shown in Fig.R1a, when the concentrations of CMC-MV²⁺, CB[8] and Azo-PLLA are fixed at 0.5, 0.25 and 0.5 mg mL⁻¹, the surface coverage reaches to $\sim 100\%$ in 600 s. Therefore, 600 s can be estimated as the threshold value for arrested coalescence and, when the assembly time is greater than 600 s, partial fusion behavior can be observed (Figure R1b). We note that, the partial fusion behavior can also be observed when the assembly time is slightly less than 600 s, e.g., 400 s. However, in this case, the coalescence of two droplets cannot

be fully arrested. By varying the concentration of CMC-MV²⁺, CB[8] or Azo-PLLA, the assembly time it takes to saturate the surface can be effectively tuned, leading to the adjustment of the threshold value (Fig.R2).

We have added a brief discussion in the manuscript as follows: “This partial fusion behavior can be observed once the surface coverage of CB[8] surfactants reaches to 100% (the threshold time for partial fusion can be estimated at 600 s at the fixed concentrations of CMC-MV²⁺, CB[8] and Azo-PLLA shown in Fig.4a,b), and after that the behavior is independent of the assembly time (Fig.4b and Fig.S14). By varying the concentration of CMC-MV²⁺, CB[8] and Azo-PLLA, the threshold time can be effectively tuned (Fig.S15)”, and added Fig.R1 and Fig.R2 in the revised supporting information as Fig.S14 and Fig.S15.

We also quantitatively investigated the effect of droplet volume on the partial fusion of the droplets. As shown in Fig.R5, with the droplet volume ranging from 2.5 to 20 μL , the partial fusion can always be observed, indicating this behavior is independent of the droplet volume. We have added a brief discussion in the revised manuscript and added Fig.R5 in the supporting information as Fig.S25.

Fig.R5. Droplet networks constructed by droplet with different volume: (a) 2.5 μL ; (b) 5.0 μL ; (c) 10.0 μL ; (d) 20.0 μL .

4. A more detailed discussion on the redox responsiveness to control droplet fusion should be provided.

Response: We appreciate the suggestion from the reviewer, and have added a detailed discussion on the redox and guest-competitive responsiveness in the revised manuscript as follows: “Due to the dynamic nature of CB[8] surfactants, the full coalescence of droplets can be triggered by injecting a small amount of aqueous solution containing a reductant ($\text{Na}_2\text{S}_2\text{O}_4$) or competitive guest (AdH) into the droplets. As discussed above, CB[8] surfactants disassemble at the interface, significantly reducing the binding energy of interfacial assembly and, driven by the reduction in the interfacial tension, the triangular droplet network rapidly becomes spherical, losing its structured shape”.

REVIEWER COMMENTS

Reviewer #1 (Remarks to the Author):

- The authors have clarified that 2 hours is not necessary for successful droplet adhesion, and have added new figures describing surface coverage. This was one of my primary concerns and the authors have addressed this point.

- o Figure 4b is still concerning – what is the distinction between 1 h and 2 h in the presented results, assuming both have full surface coverage? If the focus of this figure is demonstrating that there is no difference between the 1 hour and 2 hour cases this should be more clearly stated in the caption. What is the significance of these two panels?

- The experimental methodology presented in the manuscript has been improved and includes discussions on all pertinent experiments, including characterization of the viscoelastic shell.

- The authors defend the lack of characterization stating this is an initial attempt and future work will emphasize adaptation in devices. Multiple reviewers have raised this concern – while the presented results are novel, the lack of detail in their interpretation and characterization will be an impediment to future adoption and optimization. Given the impact of the selected venue, the publication is premature.

- o As a minimum, some analysis must be performed on the reliability and repeatability of the applications prior to publication. The demonstrations are presented entirely as pictures with colored dyes, and only one example is provided for each case with no statistical analysis. Quantifying the change in intensity or color of the droplets with respect to time then comparing the trends across multiple experiments to characterize the reliability is a necessary step prior to publication. Figure 6c is the most straightforward candidate for this, as it is a simple diffusive chain. Alternatively, figure 5f may be a better option as it involves chemical reactions within the droplets.

Reviewer #2 (Remarks to the Author):

The authors have performed a thorough revision of the manuscript and taken seriously all of the provided reviewers comments.

I asked previously about thresholds of arrest based on surface coverage and the response was helpful and informative, enhancing the revised work.

I also asked that the method of determining the elastic modulus of the drops be shared and included and that has been done.

Given the well-handled revisions and attention to all of the comments, I am happy to recommend publication of the revised manuscript.

Reviewer #3 (Remarks to the Author):

After revision, the manuscript has been much improved. Accordingly, I am pleased to recommend publication of this work in Nature Communications.

Manuscript ID: NCOMMS-23-36167A

Title: Reconfigurable Droplet Networks

Authors: Shuyi Sun, Shuailong Li, Weixiao Feng, Jiaqiu Luo, Thomas P. Russell* and Shaowei Shi*

Dear editor:

We would like to thank the reviewers for their detailed feedback to improve our submission. We have considered each comment carefully and revised our manuscript to address the issues raised.

Responses to Reviewer #1

Comments:

1. The authors have clarified that 2 hours is not necessary for successful droplet adhesion, and have added new figures describing surface coverage. This was one of my primary concerns and the authors have addressed this point.

Response: We appreciate the positive response from the reviewer.

2. Figure 4b is still concerning – what is the distinction between 1 h and 2 h in the presented results, assuming both have full surface coverage? If the focus of this figure is demonstrating that there is no difference between the 1 hour and 2 hour cases this should be more clearly stated in the caption. What is the significance of these two panels?

Response: We appreciate the comment from the reviewer. There is no difference between the 1 h and 2 h cases, as stated it in the caption of Fig.4b, “Optical images showing the process of contact and squeeze of two droplets (a) in different direction and (b) at different assembly time; In all above cases, the partial fusion behavior of two droplets can be observed.”

We discuss the significance in the manuscript, “This partial fusion behavior can be observed once the surface coverage of CB[8] surfactants reaches to 100% (the threshold time for partial fusion can be estimated at 600 s at the fixed concentrations of CMC-MV²⁺, CB[8] and Azo-PLLA shown in Fig.4a,b), and after that the behavior is independent of the assembly time (Fig.4b and Supplementary Fig.14). This result also indicates that the structure of interfacial assembly remains unchanged once CB[8] surfactants saturate the interface, which is in agreement with the AFM results in Supplementary Fig.10”.

3. The experimental methodology presented in the manuscript has been improved and includes discussions on all pertinent experiments, including characterization of the viscoelastic shell.

Response: We appreciate the positive response from the reviewer.

4. The authors defend the lack of characterization stating this is an initial attempt and future work will emphasize adaptation in devices. Multiple reviewers have raised this concern – while the presented results are novel, the lack of detail in their interpretation and characterization will be an impediment to future adoption and optimization. Given the impact of the selected venue, the publication is premature.

Response: We appreciate the comment from the reviewer. We fully understand the concern from all reviewers and have tried our best to provide quantitative characterizations in the last round of revision, which have been shown in Supplementary Fig.10, 14, 15, 16 and 28. The other two reviewers were satisfied with our revisions. In this round of revision, we have performed multiple experiments to confirm the reliability and repeatability of our results. Also, we have provided another quantitative characterization to monitor the change in intensity or color of the droplets as a function of time. Detailed data have been provided below (response to comment No. 5) and we hope these results suffice.

5. As a minimum, some analysis must be performed on the reliability and repeatability of the applications prior to publication. The demonstrations are presented entirely as pictures with colored dyes, and only one example is provided for each case with no statistical analysis. Quantifying the change in intensity or color of the droplets with respect to time then comparing the trends across multiple experiments to characterize the reliability is a necessary step prior to publication. Figure 6c is the most straightforward candidate for this, as it is a simple diffusive chain. Alternatively, figure 5f may be a better option as it involves chemical reactions within the droplets.

Response: We appreciate the comments from the reviewer and have repeated the experiments in Fig.4f and Fig.5. For each case, three parallel experiments were performed. As shown in Fig.R1-R4, similar trends across multiple experiments are obtained, confirming the reliability and repeatability of the applications. We have added Fig.R1-R4 in the revised supporting information as Supplementary Fig.24, 29, 30, 33.

Fig.R1. Repeatability of the experiment on the diffusion of dye within the droplet networks shown in Fig.4f.

Fig.R2. Repeatability of experiment on the chromogenic reaction for $\text{Fe}^{3+} + 3\text{SCN}^- = \text{Fe}(\text{SCN})_3$ within the droplet networks shown in Fig.5a-b.

Fig.R3. Repeatability of the experiment on the synthesis of ZIF-67 within the droplet networks shown in Fig.5c-d.

Fig.R4. Repeatability of the experiments on the cascade enzymatic reaction within the droplet networks shown in Fig.5e-f.

To quantify the change in intensity or color of the droplets with respect to time, we convert color images of Fig.R1 to grayscale images and plot the droplet profiles with gray values. As shown in Fig.R5 and Fig.R6, for three time-dependent experiments, the gray value of the left droplet increases, while the gray value of the right droplet decreases. We have stated this result in the revised manuscript, “By converting the color images to grayscale images and plotting the droplet profiles with gray values, time evolution of the diffusion process can be quantified”. We have also added Fig.R5-R6 in the revised supporting information as Supplementary Fig.25-26.

Fig.R5. Grayscale images of Fig.R1.

Fig.R6. Time evolution of the dye diffusion within the droplet networks by plotting the droplet profiles (Fig.R5) with gray values. The gray value is tracked from the middle of the grayscale image.

Moreover, we note that in Fig.6c, the dye diffusion within the droplet network channel is not a spontaneous diffusion process but controlled by a syringe pump, the fluid is infused through the inlet of the channel and then extracted from the outlet at a fixed flow rate (Fig.R7). We have added Fig.R7 in the revised supporting information as Supplementary Fig.34.

Fig.R7. Photograph of the continuous flow system based on the droplet network channel.

Responses to Reviewer #2

Comments:

The authors have performed a thorough revision of the manuscript and taken seriously all of the provided reviewers comments.

I asked previously about thresholds of arrest based on surface coverage and the response was helpful and informative, enhancing the revised work.

I also asked that the method of determining the elastic modulus of the drops be shared and included and that has been done.

Given the well-handled revisions and attention to all of the comments, I am happy to recommend publication of the revised manuscript.

Response: We appreciate the positive response from the reviewer.

Responses to Reviewer #3

Comments:

After revision, the manuscript has been much improved. Accordingly, I am pleased to recommend publication of this work in Nature Communications.

Response: We appreciate the positive response from the reviewer.

REVIEWERS' COMMENTS

Reviewer #1 (Remarks to the Author):

My only remaining concern was demonstrating reliability/repeatability of the described research. The authors have satisfactorily addressed this concern through a series of images in the supplementary information. I have no further recommendations prior to publication.